# Studying the Antifungal Effects of *Ageratina adenophora* (Sprengel) R. King and H. Robinson (=*Eupatorium adenophorum* Sprengel) as a Bio-Fumigant Plant Alone and in Combination with Biochar Against *Pythium aphanidermatum* (Edson) Fitz

**DOI:** 10.3390/plants13243511

**Published:** 2024-12-16

**Authors:** Shiva Parsiaaref, Aocheng Cao, Yuan Li, Asgar Ebadollahi, Ghasem Parmoon, Jalal Gholamnezhad, Qiuxia Wang, Dongdong Yan, Wensheng Fang, Zhaoxin Song, Xianli Wang, Min Zhang

**Affiliations:** 1Institute of Plant Protection, Chinese Academy of Agricultural Sciences, Beijing 100193, China; shiva.aref1391@gmail.com (S.P.); aochengcao@ippcaas.cn (A.C.); qxwang@ippcaas.cn (Q.W.); yandongdong@caas.cn (D.Y.); fangwensheng@caas.cn (W.F.); songshuting_123@163.com (Z.S.); 82101225339@caas.cn (M.Z.); 2State Key Laboratory for Biology of Plant Disease and Insect Pests, Beijing 100193, China; 3Department of Plant Sciences, Moghan College of Agriculture and Natural Resources, University of Mohaghegh Ardabili, Ardabil 5697194781, Iran; ebadollahi@uma.ac.ir; 4Sugar Beet Research Department, Kermanshah Agricultural and Natural Resources Research and Education Center, AREEO, Kermanshah 671451661, Iran; ghasem.parmoon@gmail.com; 5Department of Horticultural Sciences, Faculty of Agriculture & Natural Resources, Ardakan University, Ardakan 9549189518, Iran; jgholamnezhad@ardakan.ac.ir; 6Institute for Agro-Food Standards and Testing Technology, Shanghai Academy of Agricultural Science, Shanghai 201106, China; wangxianli@saas.sh.cn

**Keywords:** *Ageratina adenophora*, *Pythium aphanidermatum*, biochar, antifungal, main compounds

## Abstract

*Pythium* spp. are soil-borne pathogens that cause damping-off and root rot diseases in many plant species such as cucumber. In the current study, the effect of dried roots–stems and leaves of *Ageratina adenophora* (Sprengel) R. King and H. Robinson (=*Eupatorium adenophorum* Sprengel) alone and in combination with pyrogenic biomass biochar to control *Pythium aphanidermatum* (Edson) Fitz was assessed. In four treatments of leaves, roots–stems, leaves + biochar, and roots–stems + biochar, it was observed that the treatment with leaves at an E_max_ (maximal effective concentration on control fungi) of 79 g/kg of soil had the most antifungal effect on *P. aphanidermatum*. Also, the C_max_ (the highest level of control) increased with time and reached 82.4% and 71% on days 30 and 60, respectively. The highest cucumber fresh fruit weight and the highest height of the stems in the greenhouse were observed in leaf treatment of *A. adenophora*. Biochar did not have any remarkable controlling effect on *P. aphanidermatum*, and its population increased. The main compounds extracted from the dried leaves and roots–stems of *A. adenophora*, including α-pinene, nonanone, hexahydronaphthalene, 3-undecanone, muurolene, and heneicosane, had antifungal properties. We concluded that the leaves of *A. adenophora* have the potential to be used as a bio-fumigant for *P. aphanidermatum* management.

## 1. Introduction

The genus *Pythium*, belonging to the Pythaceae family and Pronosporales order, was described and named by Pringsheim in 1858 [1,2]. *Pythium* is a microscopic fungus, commonly known as a facultative parasite, and it is categorized in the class of oomycetes in most classifications [3,4,5]. Pre-damping-off and post-damping-off caused by *Pythium* species are responsible for more than 60% of deaths of germinated seedlings in fields [6,7], which is influenced by temperature, soil humidity, and rainfall [8]. Root and ring rot are common diseases of vegetables and can spread globally. *Pythium aphanidermatum* (Edson) Fitz is one of the main causes of root and ring rot in the world [2]. *P. aphanidermatum* causes diseases especially in vegetables belonging to family Cucurbitaceae [9,10]. *Pythium* species grow in soil under different temperature conditions. *Pythium* species grow in soil under different temperature conditions and *P. aphanidermatum* species grows in temperatures from 25 to 30 °C [11,12]. In plants infested with *P. aphanidermatum*, the death of tissue of the root system happens first, the inhibition of root longitudinal growth then occurs, the inhibition of plant growth occurs next, and, finally, the death of the plant occurs [13].

The synthetic fumigant pesticide methyl bromide is prohibited in most countries due to its detrimental side effects, including carcinogenic properties and depletion of the ozone layer [14]. Carbon bisulfide and chloropicrin are considered the most effective pesticide chemicals. They are used to eliminate plant pests, weeds, and plant diseases, whose causes include viruses, bacteria, fungi, and nematodes, in greenhouses and in field conditions by turning into a gas to penetrate soil particles. These fumigants are non-specific and highly hazardous for humans and the environment [14]. Alternative methods of controlling soil-borne diseases should be studied and evaluated in order to eliminate the use of methyl bromide and other detrimental fungicides [15]. The increase in pesticide production has led to more and more research being conducted on green pesticides from eco-friendly agents, natural organic soil amendments, organic waste, green manure, bio-fumigation products, compost, and essential oils. Non-toxic, non-residual, and highly degradable organic fumigants can be used as environmentally friendly alternatives to chemical pesticides to manage pests and plant diseases caused by soil pathogens [14]. Bio-fumigation is a biological control method that involves the use of volatile compounds from natural sources to inhibit a wide range of pests and pathogens and to control fungal decay [16,17]. This new approach to disease control has several potential advantages [18]. The volatile compounds used for fumigation are often harmless natural ingredients, and they can inhibit pathogenic microorganisms without direct contact with the crop [19]. Also, the continuous and slow release of active ingredients has a long-lasting inhibitory effect on soil pathogens [20]. Of course, the efficiency of fumigation largely depends on the used materials [21]. Therefore, it is very important to choose safe and effective fumigants [17].

*Eupatorium* is a flowering plant from the Asteraceae family [22]. Some species of *Eupatorium* are used to cure various diseases in traditional medicine [23,24]. *Eupatorium adenophorum* species [Syn. *Ageratina adenophora* (Spreng.) R.M. King and H. Rob.] is used as an antimicrobial plant in traditional medicine [25,26]. Although native to Mexico, it was first observed in China in the 20th century [26,27]. The dried roots–stems of *E. adenophorum* Spreng used as a bio-fumigant plant demonstrated control effects on nematode *Meloidogyne incognita* in a cucumber cultivar (Jinyou 35) in a greenhouse in China [24].

Another alternative disease management strategy involves the use of compost and biochar [28,29]. Biochar (pyrogenic biomass derived from a product) is used in soil is due to its long-term carbon sequestration potential to reduce climate change [29]. Because biochar (BC) affects and improves the physicochemical properties of soil, soil fertility, and plant growth, it can be used as a soil amendment [30,31]. Biochar is a porous pyrolytic solid product obtained from organic waste such as wood chips and any garden waste [32]. A synergic effect of biochar and compost has been reported to improve soil fertility, plant growth, and beneficial microbial activities in the rhizosphere [33,34]. Several studies have reported the capacity of biochar to suppress plant diseases [35]. Biochar has a direct effect on the effectiveness and environmental fate of pesticides by absorbing them [36]. Investigations have shown that the use of biochar as an amendment in agricultural soils reduces the microbial soil degradation caused by pesticides [37,38,39]. As pesticides reduce the bioavailability of nutrients in soil, biochar can also affect the fate of metabolites [31].

Our research had the following aims: (1) Investigate the fungicidal effects of five different concentrations of dried leaves and roots–stems of *A. adenophora* against *P. aphanidermatum* over 7, 14, 21, 30, and 60 days. (2) Study the effect of biochar combined with the dried leaves and roots–stems of *A. adenophora* in terms of *P. aphanidermatum* control. (3) Assess the impact of the dried leaves and roots–stems of *A. adenophora* on the stem height and fresh fruit weight of cucumber in greenhouses. (4) Identify the main compounds extracted from the dried leaves and roots–stems of *A. adenophora*.

## 2. Results

### 2.1. In Vitro Antifungal Effect

The results showed that the *A. adenophora* leaf and roots–stems concentration, use of biochar, and time had different effects on *P. aphanidermatum*. The interaction effects of biochar × time, biochar × concentration, and biochar × time × concentration were also significant (Table 1). The results of the comparison of the means indicated that the use of the roots–stems did not control *P. aphanidermatum* and caused a 14% increase in the fungus population. Also, the use of biochar caused an increase in the population of *P. aphanidermatum* of up to 17.3% (Table 1). The results of the comparison of the different times also showed that the use of the roots–stems increased the population of *P. aphanidermatum* at days 7, 14, and 21 (−49.5%, −40.1%, and −42.8%, respectively), but at days 30 and 60, it demonstrated control properties and also reduced the population of *P. aphanidermatum* by 35.6% and 17.7%, respectively (Table 1). The comparison of the different roots–stems concentrations also showed that all the concentrations of roots–stems increased the population of *P. aphanidermatum*, while at concentrations of from 60 to 70 g/kg, the lowest increase was observed, and at a concentration of 100 g/kg, the highest increase was seen (Table 1).

The results of fitting the non-linear regression models on the control of *P. aphanidermatum* under the influence of time and with biochar are shown in Figure 1 and Table 2. In the treatment without biochar, a Gaussian model was used at all time points (R^2^ = 0.899–0.999 and RMSE = 0.52–12.67). In the treatment with biochar, at days 7, 14, and 21, a logistic model was used to fit the changes (R^2^ = 0.995–0.975 and RMSE = 9.0–2.30), and at days 30 and 60, a Gaussian model was used (R^2^ = 0.980–0.891 and RMSE = 7.97–3.50) (Table 2).

According to the C_min_ parameters, which indicate the minimum control of the fungi, in the treatments without biochar on day 7, the minimum control of *P. aphanidermatum* at a concentration of 50 g/kg of roots–stems was −30%. This indicated that the *P. aphanidermatum* population was not controlled and also increased at this time. In addition, even on days 14 and 21, the increase in population reached −67.9%. However, C_min_ reached 15.8 and 5.6% at days 30 and 60. This shows that time had a role in the effectiveness of the bio-fumigant and increased its control properties at the minimum concentration. According to the C_max_ parameter, which indicates the maximum control of the fungi, on days 30 and 60, using the roots–stems of *A. adenophora* had a controlling effect and caused the control of 66.3% and 22.9% of the *P. aphanidermatum*, respectively. However, on days 7, 14, and 21, the bio-fumigant plant at the maximum concentration increased the population of *P. aphanidermatum*, and at C_max_, the changes were of −27%, −2.6%, and −27%, respectively (Table 2). According to the E_max_ parameter, which indicates the maximal effective concentration on the control of the fungi, it was found that the roots–stems of *A. adenophora* at a concentration of 69.7 g/kg of soil caused the highest effect in terms of the control of *P. aphanidermatum*, and at concentrations higher than this, the controlling properties reduced (Table 2).

The results also showed that in the treatment with biochar, with the passage of time from days 7 to 21, C_min_ decreased from −84.1% to −100.4%, and C_max_ increased from −21.4% to 3.7%, which shows the role of the time parameter in the effectiveness of this plant. The E_50_ parameter, i.e., the half-maximal effective concentration on the control of the fungi, from days 7 to 21 at concentrations of from 66.3 to 76.4 g/kg of roots–stems from *A. adenophora*, the population of *P. aphanidermatum* increased to 50%, which indicates a decrease in the controlling effect of the roots–stems (Table 2). At day 30 and day 60, the treatment with biochar was weaker than the treatment without biochar in terms of controlling *P. aphanidermatum*; however, in comparison with days 7 to 21, its controlling ability was better. At day 30 and day 60, C_max_ was 52.3% and 47.1%, respectively, and E_max_ was observed at concentrations of 67.4 and 66.3 g/kg of roots–stems (Table 2).

The results of the study on *P. aphanidermatum* control by *A. adenophora* dried leaves showed that the different concentrations, biochar, and times had different effects on *P. aphanidermatum* control. The interaction effects of biochar × time, biochar × concentration, and biochar × time × concentration were also significant (Table 1). The results of the comparison of the means showed that the dried leaves could control *P. aphanidermatum* and caused a 9% decrease in its population. Also, the use of biochar reduced the control properties of the dried leaves, and the population of *P. aphanidermatum* increased by up to 16.7% (Table 1). The results of the time comparison showed that in the treatment with dried leaves on days 7 and 14, the population of *P. aphanidermatum* increased, but on days 21, 30, and 60, there was a control effect of 2.6%, 32.2%, and 22.3%, respectively (Table 1). The comparison of the concentration of the dried leaves showed that control of *P. aphanidermatum* was only seen at concentrations of 70 and 80 g/kg, but at lower and higher concentrations, the population of *P. aphanidermatum* increased (Table 1).

The results of fitting the non-linear regression models of *P. aphanidermatum* control under the influence of time, concentration, and biochar, as shown in Figure 2 and Table 3, showed that in both treatments (with biochar and without biochar), Gaussian models were used at all the investigated times: R^2^ = 0.997–0.828 and RMSE = 1.7–14.6) (Table 3).

In the treatment without biochar on day 30, at a dried leaves concentration of 50 g/kg, C_min_ was 48.5%, indicating proper *P. aphanidermatum* population control at this time. Also, by decreasing or increasing the time, the population control of *P. aphanidermatum* either decreased or increased. On day 21, the dried leaves of *A. adenophora* reduced the population *P. aphanidermatum* (C_max_) to 61.8%, and on days 30 and 60, this control reached 82.4% and 71.0% (Table 3). The E_max_ of the dried leaves on days 21, 30, and 60 (73.1, 79, and 74.7 g/kg of soil, respectively) was lower than the E_max_ at days 7 and 14 (81.5 and 87.4 g/kg of soil), which indicates an increase in the control effect of the dried leaves with the passage of time (Table 3).

The results also showed that in the treatment with biochar, although C_min_ increased over time, the population of *P. aphanidermatum* was high at the minimum concentration of dried leaves. In this treatment, C_max_ reached 27.3% and 13.1% on days 30 and 60, showing the role of the time parameter in terms of biochar’s effectiveness. Also, the E_max_ parameter was in the range of from 63.4 to 83.5 g/kg of soil, and on days 30 and 60, it was seen at a minimum concentration of 63.9 and 63.4 g/kg of soil, showing that the control increased with the passage of time (Table 3).

### 2.2. In Vivo Experiments

In the greenhouse experiment, the cucumber plant height and fresh fruit weight at a concentration of 70 g/kg of soil from the dried leaves and roots–stems of *A. adenophora* and their combination with biochar were investigated and compared with a control. The concentration of *A. adenophora* had a significant controlling effect (*p* < 0.01) on *P. aphanidermatum* in the greenhouse at 25–28 °C.

The control percentage of *P. aphanidermatum* was studied from July to November. In the treatment with dried leaves at a concentration of 70 g/kg of soil, the cucumber plant height (on average 184 cm) was significantly taller than with the other three treatments and the control. The fresh fruit weight (on average, 5.89 kg/m^2^) was also higher in this treatment than with the other treatments and the control. In examining both the plant height and fresh fruit weight, it was observed that the treatment with roots–stems and the treatment with roots–stems + biochar were not significantly different and were placed in the same group. However, the treatment with leaves and the treatment with leaves + biochar were significantly different. All four treatments were significantly different in comparison with the control (Figure 3).

### 2.3. Chemical Compounds Extracted

In the GC-MS analysis of *A. adenophora*, about 26 main compounds from the dried roots–stems and 24 main compounds from the dried leaves were identified (Appendix A). D-limonene, undecane, muurolene, and heneicosane were the most important main compounds identified from the roots–stems. In this analysis, over the time, the concentration of d-limonene decreased, but that of the other compounds, undecane, muurolene, and heneicosane, increased, and their highest concentrations were observed after day 21 (Appendix A). α-Pinene, nonanone, and hexahydro naphthalene were the most important main compounds identified from the leaves, and their highest concentrations, 18.9, 10.9, and 6.2 mg/g, respectively, were observed at day 21 (Appendix A).

### 2.4. Correlation and Clustering

The correlation between the results of the compounds identified from the roots–stems and leaves and the control of the *P. aphanidermatum* population was studied, and the results are shown in Figure 4 and Figure 5. The main compounds, norbornane, naphthalene, hexanoic acid, adamantane, muurolene, and octocrylene, from the roots–stems had the highest correlation in terms of the control of *P. aphanidermatum* and were classified into one group (Figure 4). The main compounds, dimethyl-1,6-octadiene, cyclopentasiloxane, α-pinene, methyl-2-methylene, and hexahydro naphthalene, from the leaves had the highest correlation in terms of the control of *P. aphanidermatum* and were classified into a different group (Figure 5).

## 3. Discussion

The laboratory studies showed that *P. aphanidermatum* was effectively controlled at concentrations of 70 and 80 g/kg of dried leaves of *A. adenophora*. In this treatment, on day 21, C_max_ was 61.8%, which increased to 82.4 and 71.0% within 30 and 60 days. However, in the treatment with dried leaves + biochar, the C_max_ parameter on days 30 and 60 was 27.3 and 13.1%, indicating the effectiveness of the time parameter when using biochar. In the treatment with dried leaves, on days 21, 30, and 60, E_max_ was observed at concentrations of 73.1, 79, and 74.7 g/kg of soil, in which *P. aphanidermatum* control was seen at 61.8%, 82.4%, and 71%, respectively. Concurrently, in the treatment with dried leaves + biochar, on days 30 and 60, E_max_ was observed at concentrations of 63.9 and 63.4 g/kg of soil, in which 27.3% and 13.1% control of *P. aphanidermatum* was seen, respectively (Table 3). The potential of *A. adenophora* and biochar in the management of *Pythium* fungi was evaluated in recent studies. Oil extracted from the leaves of *E. adenophorum* was studied in terms of controlling the growth of *P. myriotylum* over seven days, and it was found that the mycelium growth decreased by increasing the concentration of oil. The mycelial growth was completely inhibited at concentrations of 100 and 120 μg/mL after seven days of incubation [40]. To reduce the root rot disease of Poinsettia (*Euphorbia pulcherrima*) caused by *Pythium aphanidermatum*, in a greenhouse experiment, mixed hardwood biochar, which was used instead of peat moss (commercial substrate), was studied. Hardwood biochar was mixed with commercial substrate at 0%, 20%, and 40% [CS100 (control), HB20, and HB40, respectively, by volume]. In the presence of the pathogen, the plants grown in the HB20 treatment had significantly higher shoot dry weights but lower disease severity and disease incidence than the control [41].

In the present study, the effects of the fungicidal properties of dried roots–stems of *A. adenophora* on controlling *P. aphanidermatum* were studied for the first time. In the laboratory study of the roots–stems, C_max_ was 66.3% and 22.9% on days 30 and 60. In the treatment with roots–stems + biochar, C_max_ was 52.3% and 47.1% at days 30 and 60, respectively. In the treatment with roots–stems, on days 30 and 60, E_max_ was obtained at a concentration of 69.7 g/kg of soil on both days, and the level of *P. aphanidermatum* control reached 66.3% and 22.9%, respectively. However, in the treatment with roots–stems + biochar, on days 30 and 60, E_Max_ was seen at concentrations of 67.4 and 66.3 g/kg of soil, indicating 52.3% and 47.1% control of *P. aphanidermatum*, respectively. In line with our findings, in order to control the damping-off of cucumber caused by *P. aphanidermatum*, canola residues in increasing amounts (30 to 120 tons/ha) were used to treat infested plots and incorporated into the soil. The damping-off level and seedling inoculation decreased with the increase in the canola residue levels. Bio-fumigation of soil with canola residues was an effective tool for managing cucumber damping-off [42].

Biochar can increase plant growth and reduce diseases. A biochar-amended growth medium was preconditioned with fertilization before planting to enhance the structure and activity of the native microbial community. Cucumber plant performance and resistance to damping-off caused by *P. aphanidermatum* were investigated. Preconditioning increased the efficiency of the biochar in improving plant performance and suppressing soil-borne diseases by enriching the environment with beneficial soil microorganisms, increasing the microbial and fungal diversity and activity, and removing compounds of biochar that are toxic to plants. The findings showed that pre-aeration should be included as an important step during biochar application in soil and in soilless environments [43]. In the comparison of the four treatments, i.e., leaves, roots–stems, leaves+ biochar, and roots–stems + biochar, it was observed that the treatment with leaves with C_max_ = 82.4% demonstrated the highest control of *P. aphanidermatum*, but adding biochar to the treatments at certain times and concentrations increased the *P. aphanidermatum* population. However, it was noticed that the treatment with roots–stems + biochar with C_max_ = 52.3% on day 30 had a higher control effect against *P. aphanidermatum* compared to the treatment with leaves + biochar with C_max_ = 27.3% on day 30.

In the greenhouse study, the effects of bio-fumigants and fungicides from dried roots–stems, dried roots–stems+ biochar, dried leaves, and dried leaves + biochar from *A. adenophora* were studied in terms of the control of *P. aphanidermatum*. In the comparison of the four treatments, the highest plant height and fresh fruit weight were as high as 184 cm and 5.89 kg/m^2^ in the treatment with dried leaves. On the other hand, the lowest plant height and fresh fruit weight were observed in the control, which were 136 cm and 16.4 kg/m^2^, respectively. Finally, it should be noted that cucumber (cv “Jinyou 35”) is susceptible to *P. aphanidermatum*, so the fungi can cause a decrease in crop yields in contaminated greenhouses. Solarization and bio-fumigation were studied to manage damping-off disease caused by *P. aphanidermatum* in cucumber greenhouses. Solarization and bio-fumigation (solarization following organic amendment of the soil) both reduced the level of *P. aphanidermatum* inoculation in the soil compared to the control group. Bio-fumigation and solarization both increased the plant height and stem diameter [15].

The main compounds of the dried roots–stems and leaves from *A. adenophora* were identified by the SPME method. The most important leaf compounds that have *P. aphanidermatum* control properties included α-pinene, nonanone, and 6,7-dimethyl-1,2,3,5,8,8a-hexahydronaphthalene. The main compound, found in plants with significant antimicrobial activity, was α-pinene. A study was conducted to evaluate the antifungal activity of α-pinene on *Candida* species isolated from otomycosis. α-Pinene showed significant antifungal activity with greater inhibitory and fungicidal activity against *C. parapsilosis* and proved to be effective in inhibiting fungal structures, such as pseudo-hyphae, and in promoting a significant reduction in blastoconidia [44]. The efficacy of the antimicrobial activity of active films against *Botrytis cinerea* was evaluated in vitro based on a complex incorporated with β-cyclodextrin, 2-nonanone, and two polymer matrices (polylactic acid and low-density polyethylene). The microbiological analysis showed high effectiveness in terms of reducing the growth of *B. cinerea*. The active films developed in the study were able to inhibit the growth of the pathogenic fungus *B. cinerea* in different experimental conditions [45]. In another study, the antifungal effect of citronella oil on *Aspergillus niger* ATCC 16404 was investigated. The experimental results showed that the citronella oil had strong antifungal activity. A total of 125 (*v*/*v*) and 0.25% (*v*/*v*) citronella oil inhibited the growth of 5 × 10^5^ spores/mL conidium separately for 7 and 28 days, while 0.5% (*v*/*v*) citronella oil completely killed 5 × 10^5^ spores/mL. The citronella oil caused the tough cell walls to rupture and then acted on the sporoplasm to kill the conidia. In line with the present study, the citronella oil had important compounds, including 6,7-dimethyl-1,2,3,5,8,8a-hexahydronaphthalene, at a concentration of 2.32% [46].

In the analysis of the dried roots–stems of *A. adenophora* using the SPME method, 26 main compounds were identified, in which 3-undecanone, γ-muurolene, and heneicosane, which have fungicidal and antimicrobial properties, were the most effective in controlling *P. aphanidermatum*. The antimicrobial activity of undecan-x-ones (x = 2–4) was investigated against the bacteria *Escherichia coli* and *Bacillus subtilis*, the yeast *Candida mycoderma*, and the fungus *Aspergillus niger*. The undecane-x-ones showed low antibacterial activity against Gram-positive and Gram-negative bacteria. Undecane-2-one and undecane-3-one showed high activity against *C. mycoderma*. Similar to the results of the present study, all the undecane-x-ones expressed the strongest fungicidal effect [47]. α-Pinene (4.3%), β-pinene (11.4%), limonene (6.7%), β-caryophyllene (8.2%), γ-muurolene (7.3%), bicyclogermacrene (8.0%), E-nerolidol (9.6%), and spathulenol (9.8%) were the main compounds identified from the essential oil of *Baccharis semiserrata* leaves. The antifungal activity of the essential oils was determined by the agar dilution method. Both the essential oils of leaves and twigs of *B. semiserrata* were active against *Microsporum gypseum*, *Candida albicans*, *Epidermophyton flocosum*, *Trichophyton mentagrophytes*, and *Cryptococcus neoformans*, and the activity was related to the main compounds, in line with our study [48]. The cuticular composition of the fresh leaves of teak (*Tectona grandis*. L), which is responsible for wide adaptability and antimicrobial activity, was investigated. Spectral analyses showed a long unbranched heneicosane chain with a mol. wt. of 296. This saturated hydrocarbon forms a continuous layer on leaf surfaces that acts as a physical barrier to microorganisms and also has a strong defensive activity against a number of pathogens, indicating its role in the defense mechanisms of the plant. The antifungal activity of heneicosane on each of the microbial species was significant [49].

The results obtained in the present research and in the previous studies indicate that *A. adenophora* has a high potential to control *P. aphanidermatum* and that it can be used as a bio-fumigant plant in *Pythium*-contaminated greenhouses as a biological control. Also, the dried leaves of *A. adenophora* were stronger than the dried roots–stems, and it is recommended to use the leaves as a biological fungicide to control *P. aphanidermatum*. α-Pinene, nonanone, and 6,7-dimethyl-1,2,3,5,8,8a-hexahydronaphthalene were the main compounds of the dried leaves from *A. adenophora* that had fungicidal properties, and it is assumed that these compounds were the main factors in controlling *P. aphanidermatum*.

## 4. Materials and Methods

### 4.1. Bio-Fumigant Plant

Different parts of *A. adenophora* (roots, stems, and leaves) were collected from the uncultivated areas of Xichang City, Sichuan Province, China. *Ageratina adenophora* was identified by Prof. A. C. Cao (Institute of Plant Protection, Chinese Academy of Agricultural Sciences, Beijing, China). A voucher specimen (No. 20060712) was deposited at the College of Chemistry, Beijing Normal University, Beijing 100875, People’s Republic of China [50]. In order to dry the roots, stems, and leaves of *A. adenophora*, they were kept for a week in a dark and dry room at 28 °C. A grinder (Qufu Shunyang Machinery Co., Ltd., Jining, China) was used to powder the roots–stems and leaves separately. Then, they were packed in separate vacuum-packed Ziplock plastic bags and kept at a temperature of 28 °C in the dark.

### 4.2. Identification of Chemical Compounds

#### 4.2.1. Headspace Solid-Phase Microextraction (SPME)

In order to determine the fiber with the optimal absorption of fumigants, a type of SPME extraction fiber (divinylbenzene/carboxen/polydimethylsiloxane (DVB/CAR/PDMS)) (Bonna-Agela Technologies Co., Ltd., Tianjin, China) was used [51]. The sample was placed in a 20 mL headspace vial [52]. The SPME needle was passed through the septum of the vial, and the fiber was exposed to the headspace (HS) for 20 min at 65 °C to absorb the vaporized compounds. Then, the fiber was removed and placed in the injection port for 5 min to measure the desorption at 260 °C [51].

#### 4.2.2. Gas Chromatography–Mass Spectrometry (GC-MS)

The Kundu, 2016, [53] method was used to study the main compounds of *A. adenophora* dried roots–stems and leaves in a GC analysis. The treatments were studied separately on days 7, 14, 21, and 30, at a temperature of 25 °C, with three replications. A SHIMADZU GC-MS QP 2010 (SHIMADZU Co., Ltd., Kyoto, Japan), set in the Select Ion Monitoring mode, was used for gas chromatography–mass spectroscopy (GC-MS) analysis. The operating conditions were as follows: RTX-5MS capillary column (30 m long, 0.25 mm ID, 0.25 µm thick); helium carrier gas at a flow rate of 1.5 mL min^−1^; oven temperature initially at 65 °C for 1 min, then raised at 3 °C min^−1^ until 250 °C; 230 °C interface temperature; and 350 °C ion source temperature. The mass acquisition parameters were as follows: ion source, 180 °C; transfer line temperature, 70 °C. The spectra stored in the NIST spectral library were used to identify the relevant chemicals in the GC column [24].

### 4.3. In Vitro Studies

#### 4.3.1. Collection of Samples

##### Pathogen

In order to obtain isolates of *Pythium* sp., sampling of infested plants, which had symptoms such as damping-off, yellowing, and wilting, was carried out in cucumber greenhouses in Shunyi, Beijing, China. The collected samples were washed under water and then cut into 10 mm pieces. The chopped pieces were washed in sterile deionized water and dried in sterile paper towels. Based on the Jeffers and Martin 1968 [54] method, 10 mm root sections were placed onto agar plates containing PARP medium and then incubated at 25 °C in the dark. After 36 h, the recovered fungal colonies were moved on water agar (15 g of agar, 1 L of distilled water), and the hypha tip method was used for their purification. Based on the Tuite, 1969, [55] method, the morphology of the colonies was studied on CMA (extract of 60 g of ground maize, 15 g of agar, and 1 L of distilled water) and PCA (extract of 20 g of carrots and 20 g of potatoes, 15 g of agar, and 1 L of distilled water) at 25 °C. The growth rate was measured on CMA at 5, 15, 20, 25, 30, 35, 40, and 45 °C. To induce the formation of sporangia, 5 mm pieces of autoclaved grass leaf (*Poa annua* L.) were placed on PCA plates of *Pythium* at 25 °C. After 24 h, they were transferred to a Petri dish in a shallow layer of sterile water under fluorescent light. The sexual organs were studied on HSA (extract of 20 g of ground hemp seeds, 15 g of agar, and 1 L distilled water) [56]. The methods of Van der Plaats-Niterink, 1981, [2] and Dick, 1989, [57] were used to identify *Pythium* strains.

##### Pathogenicity Assays and Race Determination

In order to carry out the pathogenicity assay, pots (10 cm in diameter) containing sterile soil (1:1:1 ratio of leaf soil–sand–soil) and cucumber seedlings (cv “Jinyou 35”) (Tiancubic Seed Industry Co., Ltd., Tianjin, China) at the three–four-leaf stage were used. An inoculum containing spore suspensions (sterile water to give a spore concentration of 10^6^/mL) [58] per gram of soil was added to the soil in pots [56]. The pots were kept in the greenhouse.

#### 4.3.2. In Vitro Antifungal Effect

Random samples of the soil were taken from 20 cm below the soil from a fungi-free cucumber greenhouse in Shunyi, Beijing, China. After the soil was wetted, it was sterilized in an autoclave at 85 °C for 30 min. The results of the analysis showed that the greenhouse soil included 12.3% sand, 64.4% silt, and 23.3% clay, with an organic matter content of 33.5 g kg^−1^ soil, pH 6.5, and moisture level of 17.8% (*w*/*w*). The methods of Mao et al. [59] and Conklin, 2005, [60] were used to measure the soil pH. In a flask, 10 mL of distilled water was added to 10 g of soil, and the mixture was stirred. The soil and water mixture was kept at rest for 10 min. The soil was separated from the water using filter paper. In this method, a pH meter (Shanghai Inesa Scientific Instrument Co., Ltd., Shanghai, China) was used [61]. In order to carry out the thermo-gravimetric analysis, the different soil samples collected from 20 cm below the greenhouse soil were weighed and heated in an oven (electric constant-temperature blast-drying oven (DHG-9240A) produced by Beijing Luxi Technology Co., Ltd., Beijing, China). This method was performed within 24 h at 105 ± 5 °C [61,62].

Two types of treatments were studied in this research. The bio-fumigant effects of the dried roots–stems and leaves of *A. adenophora* on *P. aphanidermatum* were studied in the first treatment. The bio-fumigant effects of the dried roots–stems and leaves of *A. adenophora* combined with biochar were studied in the second treatment. To study the first treatment, 5 concentrations (5, 6, 7, 8, and 10 gr) of *A. adenophora* were selected. In the second treatment, 3.5 g of biochar was added to each concentration of first treatment. Sterilized greenhouse soil (100 g), sieved with a 2 mm mesh, was used for each treatment in both experiments. Different concentrations of dried roots–stems and leaves from *A. adenophora* as a bio-fumigant plant were added to the treatments. A spore suspension with a spore concentration of 10^6^ mL of *P. aphanidermatum* was used to infest the treatments [58]. The control group included sterile greenhouse soil wetted and infested with *P. aphanidermatum* without *A. adenophora*. Plastic 250 g test tubes were used for all treatments. The closed tubes were incubated five times (7, 14, 21, 30, and 60 days) at 25 °C in the dark. The treatments and controls were studied in 3 replicates. The method described by Masago, 1977, [63] was used to quantify the *P. aphanidermatum*. The fungi colonies were cultured for 3 days at 25 °C. Colony-forming units (CFUs) per g^−1^ soil were counted for the data analysis. The content of the selected media for the cultivation of *P. aphanidermatum* is reported in Table 4 [64].

### 4.4. In Vivo Assays

Two experiments were performed in two cucumber (cv “Jinyou 35”) greenhouses at a commercial property in Shunyi, Beijing, China (40°03010.4800 N; 116°5602.1200 E). Two treatment groups were studied: group one consisted of the dried roots–stems and leaves of *A. adenophora* at a concentration of 70 g/kg of soil, and group two consisted of the dried roots–stems and leaves of *A. adenophora* at a concentration of 70 g/kg of soil combined with biochar at a concentration of 35 g/kg of soil. The control consisted of soil contaminated with *P. aphanidermatum* and without any bio-fumigant. This experiment was performed at a temperature of 25–28 °C. In each greenhouse, five plots, including four treatment plots and one control plot with three replications, were designed completely randomly. In each plot, 30 pots were studied in 3 rows of 10.

Each pot contained 5 kg of sterile soil (12.3% sand, 64.4% silt, and 23.3% clay, with an organic matter content of 33.5 g kg^−1^ soil, pH 6.5, and moisture content of 17.8% (*w*/*w*)). Fifteen-day-old cucumber seedlings “cv Jinyou 35” were planted in the pots containing sterile soil. The inoculum was added as a suspension at a concentration of 10^6^ mL of spores of *P. aphanidermatum* to the soil in the pots. Seven days after inoculation, the bio-fumigant plants (dried roots–stems and leaves of *A. adenophora*), i.e., group one, and the bio-fumigant plants (dried roots–stems and leaves of *A. adenophora* combined with biochar), i.e., group two, were added to the treatment pots [58].

### 4.5. Parameter Measurement

The efficacy of controlling the fungi was calculated according to the following equation [64]:$$Y=\left(\frac{Au-At}{Au}\right)\times 100$$
where *Y* is the fungal growth inhibition rate based on the CFU count (%), *Au* is the number of fungal colonies in the control, and *At* is the number of fungal colonies in the treatment.

### 4.6. Data Analysis

All the assays were carried out in a completely randomized factorial design. Each treatment comprised three replicates. Analysis of variance (ANOVA) and normality testing of the data were conducted using the SPSS 22 software. The correlation was conducted using the R 4.3.2.1 software. For the determination of the non-linear regression trends of the mortality response of *P. aphanidermatum*, the sigma plot 12 and R 4.3.2.1 software packages were used.

The models used included the following:Logistic: Y = C_min_ + a/(1 + (−(X/E_50_)^slope^))
Gaussian: Y = C_min_+ a × exp (−0.5 × (x − E_max_)/slope)^2^)
where C_min_ is minimum control of the fungi, C_max_ (C_min_ + a) is the maximum control of the fungi, E_50_ is the half-maximal effective concentration on the control of the fungi, and E_max_ is the maximal effective concentration on the control of the fungi.

R^2^ and the root mean square error (RMSE) were applied to determine the best estimates of the parameters. R^2^ was calculated using the following formula:R^2^ = RSS/TSS
where RSS denotes the sum of squares (SS) for the regression (∑n i = 1 L − Ḹ), and SST denotes the total SS (∑n i = 1 Li − Ḹ). Li is the observed value, and Ḹ is the corresponding estimated value. In addition, the RMSE and AICc were calculated using following the formulae:
$$\mathrm{RMSE}=\sqrt{\left(1/n\right)}\sum {\left(Y_{\mathrm{obs}}-Y_{\mathrm{pred}}\right)}^{2}$$
$$AICc=n\ln\left(\frac{RSS}{n}\right)+2k+(\frac{2k(k+1)}{n-k-1})$$
where Y_obs_ denotes the observed value, Y_pred_ denotes the predicted value, and n is the number of samples [65].

## 5. Conclusions

The porous physical structure of biochar increases the water-holding capacity of soil and acts as a habitat for soil microorganisms, including the penetration of fungal hyphae. However, little is known about how soil fungi interact with biochar. In the present study, the dried leaves and roots–stems of *A. adenophora* controlled the population of *P. aphanidermatum* at a high percentage, but adding biochar to these treatments reduced the control of *P. aphanidermatum* at a low percentage. Although one of the hypotheses in this research was that the effect of the biochar would be to increase the antifungal properties of the bio-fumigant plant, it is assumed that the main compounds of the bio-fumigant plant had an effect on the performance of the biochar and its interaction with *P. aphanidermatum*. This is the first time that *A. adenophora* has been studied as a dried plant to control *P. aphanidermatum*, and more research is needed to determine the mechanism of the increase in the *P. aphanidermatum* population caused by the combination of *A. adenophora* and biochar.

## Figures and Tables

**Figure 1 plants-13-03511-f001:**
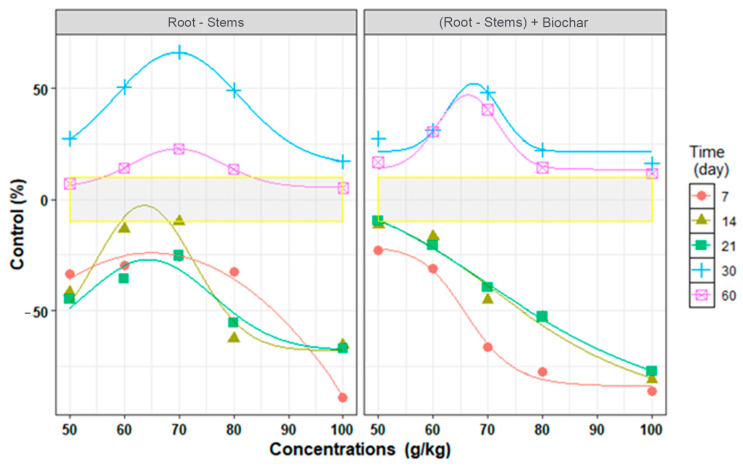
Regression relationship between in vitro control of *Pythium aphanidermatum* and different concentrations of *Ageratina adenophora* roots–stems at different times by Gaussian and Logistic models. Points are observed values, and lines are predicting values.

**Figure 2 plants-13-03511-f002:**
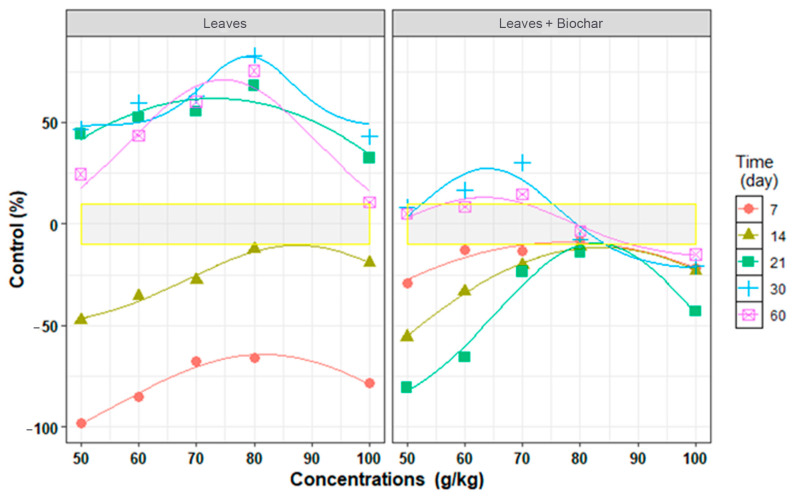
Regression relationship between in vitro control of *Pythium aphanidermatum* and different concentrations of *Ageratina adenophora* leaves at different times, shown by Gaussian models. Points are observed values, and lines are predicted values.

**Figure 3 plants-13-03511-f003:**
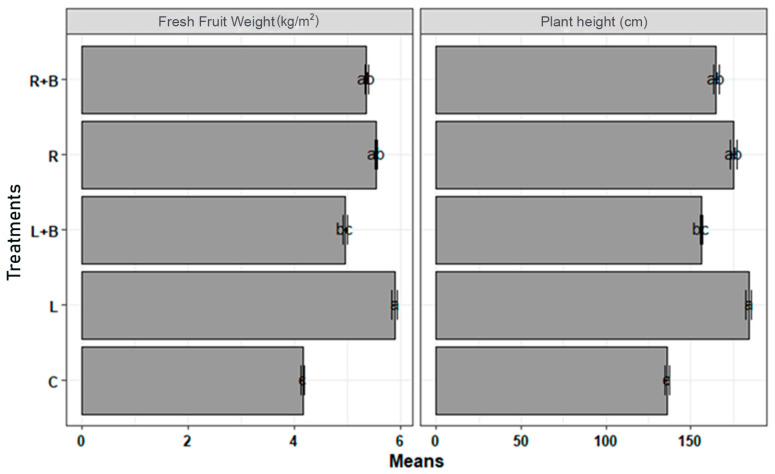
Effects of different parts of *Ageratina adenophora* and biochar on plant height and fresh fruit weight of cucumber in terms of in vivo control of *Pythium aphanidermatum*. L: leaves, R: roots–stems, B: biochar. Different letters indicate the significant differences of means in lsd test.

**Figure 4 plants-13-03511-f004:**
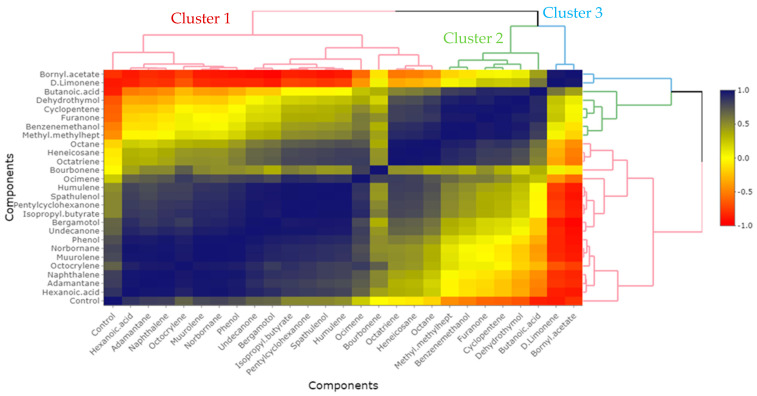
Correlation and clustering between main chemical compounds extracted from dried roots–stems of *Ageratina adenophora* with SPME over four weeks at 25 °C and control of *Pythium aphanidermatum*. Inside the graph, the dark blue color represents a correlation of +1, while the red color represents a correlation of −1. Outside the graph, red, green, and blue clusters show clusters 1, 2, and 3, respectively.

**Figure 5 plants-13-03511-f005:**
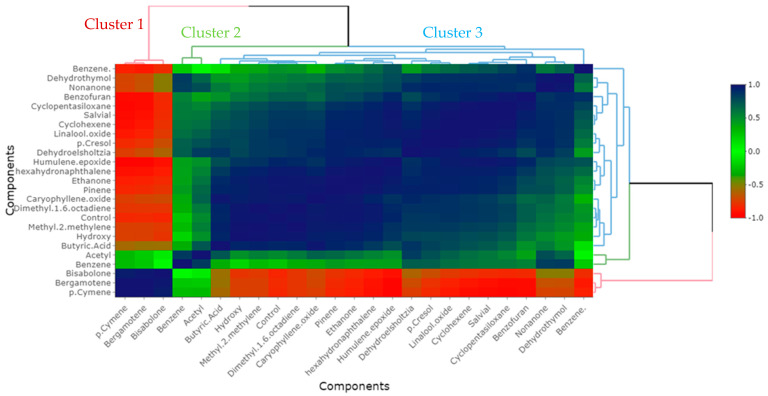
Correlation and clustering between main compounds extracted from dried leaves of *Ageratina adenophora* with SPME over four weeks at 25 °C and control of *Pythium aphanidermatum*. Inside the graph, the dark blue color represents a correlation of +1, while the red color represents a correlation of −1. Outside the graph, red, green, and blue clusters show clusters 1, 2, and 3, respectively.

**Table 1 plants-13-03511-t001:** In vitro analysis of variance in the control of *Pythium aphanidermatum* exposed to soil with dried roots–stems and leaves of *Ageratina adenophora*.

Treatments	Levels	Roots–Stems (%)	Leaves (%)
Biochar	R-S and L	−14.4 ± 8.2 a	9.2 ± 6.5 a
	(R-S + B) and (L + B)	−17.3 ± 8.4 b	−16.7 ± 5.1 b
Time (day)	7	−49.5 ± 8.5 c	−48.0 ± 9.0 d
	14	−40.1 ± 8.2 c	−28.6 ± 4.6 c
	21	−42.8 ± 67 c	2.6 ± 7.2 b
	30	35.6 ± 5.3 a	32.2 ± 7.1 a
	60	17.7 ± 3.4 b	22.3 ± 9.1 a
Concentration (g/kg)	50	−8.4 ± 8.1 ab	−18.8 ± 7.8 c
	60	−2.3 ± 9.7 a	−5.2 ± 10.1 b
	70	−3.2 ± 8.0 a	7.1 ± 9.2 a
	80	−23.8 ± 8.6 b	10.3 ± 8.8 a
	100	−41.1 ± 7.8 c	−13.8 ± 6.7 bc
F value	Biochar (B)	**	**
	Time (T)	**	**
	Concentration (C)	**	**
	B × T	**	**
	B × C	**	**
	T × C	**	**
	B × T × C	**	**
Coefficient of variation (%)		16.3	29.6

** indicates significant at 1%. Letters a–d indicate the significant differences of means in lsd test. R-S: roots–stems, L: leaves, B: biochar.

**Table 2 plants-13-03511-t002:** Estimate parameter models of regression relationship between in vitro control of *Pythium aphanidermatum* exposed to soil containing different concentrations of roots–stems from *Ageratina adenophora* at different times.

Time (days)	Treatment	Models	Estimated Parameter					R^2^	RMSE
			**C_max_**	**Slope**	**E_max_**	**E_50_**	**C_min_**		
7	R-S	G	−27.0 ± 1.9	16.9 ± 5.6	65.0 ± 2.5	-	−30.0 ± 1.9	0.987	5.95
	R-S + B	L	−21.4 ± 4.9	−15.7 ± 4.6	-	66.3 ± 1.6	−84.1 ± 6.6	0.995	4.21
14	R-S	G	−2.6 ± 6.3	9.1 ± 2.9	63.7 ± 1.9	-	−67.9 ± 12.5	0.942	12.67
	R-S + B	L	−0.9 ± 3.8	−5.6 ± 7.2	-	75.5 ± 12.3	−96.9 ± 8.2	0.975	9.00
21	R-S	G	−27.0 ± 3.8	11.7 ± 4.7	64.4 ± 3.0	-	−67.9 ± 11.4	0.899	10.41
	R-S + B	L	3.7 ± 3.6	−4.6 ± 2.2	-	76.4 ± 5.2	−100.4 ± 6.3	0.998	2.32
30	R-S	G	66.3 ± 0.5	11.4 ± 0.3	69.7 ± 0.2	-	15.8 ± 0.48	0.999	0.38
	R-S + B	G	52.3 ± 2.6	4.8 ± 5.6	67.4 ± 5.2	-	21.6 ± 5.8	0.891	7.97
60	R-S	G	22.9 ± 0.65	8.4 ± 0.4	69.7 ± 0.3	-	5.6 ± 0.45	0.999	0.52
	R-S + B	G	47.1 ± 10.0	5.5 ± 2.6	66.3 ± 1.3	-	13.7 ± 2.9	0.980	3.50

R-S: roots–stems, B: biochar, G: Gaussian, L: logistic, C_min_: minimum control of fungi, C_max_: maximum control of fungi, E_50_: half-maximal effective concentration on control of fungi, E_max_: maximal effective concentration on control of fungi, R^2^: coefficient of determination, RMSE: root mean square error.

**Table 3 plants-13-03511-t003:** Estimated parameter models of regression relationship between in vitro control of *Pythium aphanidermatum* exposed to soil containing different concentrations of leaves from *A. adenophora* at different times.

Time (days)	Treatment	Models	Estimated Parameter				R^2^	RMSE
			**C_max_**	**Slope**	**E_max_**	**C_min_**		
7	L	G	−64.2 ± 6.1	26.4 ± 7.2	81.5 ± 1.7	−131.6 ± 16.3	0.981	3.7
	L+B	G	−8.9 ± 5.8	134.7 ± 5.8	77.4 ± 0.8	−90.1 ± 5.1	0.899	5.2
14	L	G	−10.4 ± 1.8	18.6 ± 5.8	87.4 ± 1.9	−51.9 ± 10.3	0.982	3.7
	L+B	G	−11.5 ± 2.6	56.4 ± 6.2	83.5 ± 2.0	−283.3 ± 9.8	0.997	1.7
21	L	G	61.8 ± 3.6	102.0 ± 2.7	73.1 ± 3.9	−79.1 ± 6.7	0.828	11.0
	L+B	G	−9.4 ± 1.9	17.3 ± 5.4	82.7 ± 3.6	−97.2 ± 15.4	0.971	9.5
30	L	G	82.4 ± 3.6	7.7 ± 0.3	79.0 ± 5.5	48.5 ± 12.2	0.861	11.8
	L+B	G	27.3 ± 3.4	12.2 ± 5.6	63.9 ± 2.4	−22.1 ± 2.4	0.874	14.3
60	L	G	71.0 ± 11.7	16.1 ± 2.4	74.7 ± 4.2	−5.9 ± 12.4	0.922	14.6
	L+B	G	13.1 ± 2.9	14.7 ± 3.4	63.4 ± 3.4	−16.7 ± 9.4	0.916	6.7

L: leaves, B: biochar, G: Gaussian, C_min_: minimum control of fungi, C_max_: maximum control of fungi, E_max_: maximal effective concentration on control of fungi, R^2^: coefficient of determination, RMSE: root mean square error.

**Table 4 plants-13-03511-t004:** Culture medium composition (making 2 L of culture medium).

Media * Category	Ingredients	Composition	References
*Pythium aphanidermatum* (Edson) Fitz	A	Agar (34 g), glucose (40 g).	Masago [63]
	B	Pentachloronitrobenzene (PCNB) (0.15 g), Ampicillin (0.03 g), and Rifampicin (0.02 g).	

* Taking the amount of 2 L as an example: the ingredients in A were added to 1.9 L of distilled water, boiled, packed, and then sterilized; the ingredients in B were directly added to 100 mL of sterilized water and shaken.

## Data Availability

The data presented in this study are available in this manuscript.

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
