# Peer review of "Studying the Antifungal Effects of Ageratina adenophora (Sprengel) R. King and H. Robinson (=Eupatorium adenophorum Sprengel) as a Bio-Fumigant Plant Alone and in Combination with Biochar Against Pythium aphanidermatum (Edson) Fitz"

_plants, 2024, doi:10.3390/plants13243511_

Round 1
Reviewer 1 Report
Comments and Suggestions for Authors
Pythium spp. are soil-borne pathogens that cause damping–off and root rot diseases on many plant species such as cucumber. In the current study, the effect of dried root-stems and leaves of E. adenophorum alone and in combination with pyrogenic biomass biochar to control Pythium aphanidermatum (Edson) Fitz was assessed. The current name of E. adenophorum is different and no voucher specimens were included. In four treatments of leaves, root-stems, leaves + bio-char, and root-stems + biochar, it was observed that the leaves treatment at Emax (maximal effective concentration on control fungi) of 79 g/kg of soil had the most antifungal effect on P. aphanidermatum. The highest cucumber fresh fruit weight and the highest height of the stems in the greenhouse were observed in leaf treatment of E. adenophorum. Biochar did not have any remarkable controlling effect on P. aphanidermatum and its population increased. The main compounds extracted from the dried leaves and root-stems of E. adenophorum, including α-pinene, nonanone, hexahydronaphthalene, 3-undecanone, muurolene, and heneicosane, had antifungal properties. They concluded that the leaves of E. adenophorum had the potential to be used as a bio-fumigant for P. aphanidermatum management.
Introduction
Line77 King and Robinson (1987) cite Eupatorium adenophorum as a synonym of Ageratina adenophora and this is widely accepted in the botanical literature and databases online, not how it was presented in the text. Agratina adenophora does not exist.
Line 78 it is exclusively distributed throughout the world. However King and Robinson (1987) cite both A. adenophora and A. riparia as widely distributed weeds.
In the introduction there is no mention that the bio-fumigant plant used has been recorded with fungicidal effects in its leaves (due to essential oils), a reason for choosing this species, and only later in the Discussion (line 279-283) is it mentioned. This would probably explain why the leaves are more effective than the root-stems.
Line 260 (Fig. 4) and line 264 (Fig. 5) The figures are not stand alone, need to explain the clustering (PCA?) and identify the figures without needing to consult the text.
Line 387 There are no voucher specimens of the bio-fumigant plant used in the experiments and cited as deposited in a recognised herbarium, making later verification impossible.
Comments on the Quality of English LanguageThe English needs improving such as:
Line 190 Table 3 heading time should be times also in line 208.
Line 247 supplemental should be supplementary.
Author Response
Pythium spp. are soil-borne pathogens that cause damping–off and root rot diseases on many plant species such as cucumber. In the current study, the effect of dried root-stems and leaves of E. adenophorum alone and in combination with pyrogenic biomass biochar to control Pythium aphanidermatum (Edson) Fitz was assessed. The current name of E. adenophorum is different and no voucher specimens were included. In four treatments of leaves, root-stems, leaves + bio-char, and root-stems + biochar, it was observed that the leaves treatment at Emax (maximal effective concentration on control fungi) of 79 g/kg of soil had the most antifungal effect on P. aphanidermatum. The highest cucumber fresh fruit weight and the highest height of the stems in the greenhouse were observed in leaf treatment of E. adenophorum. Biochar did not have any remarkable controlling effect on P. aphanidermatum and its population increased. The main compounds extracted from the dried leaves and root-stems of E. adenophorum, including α-pinene, nonanone, hexahydronaphthalene, 3-undecanone, muurolene, and heneicosane, had antifungal properties. They concluded that the leaves of E. adenophorum had the potential to be used as a bio-fumigant for P. aphanidermatum management.
Answer: Thank you for your positive remarks about our manuscript. We will consider all comments carefully to remove any concern.
Line77: King and Robinson (1987) cite Eupatorium adenophorum as a synonym of Ageratina adenophora and this is widely accepted in the botanical literature and databases online, not how it was presented in the text. Agratina adenophora does not exist.
Answer: Thank you very much for your good comment. Unfortunately, the scientific name is written incorrectly. The scientific name was corrected in the text: “Eupatorium adenophorum species [Syn. Ageratina adenophora (Spreng.) R.M. King & H. Rob.] is used as an antimicrobial plant in traditional medicine”.
Line 78: It is exclusively distributed throughout the world. However King and Robinson (1987) cite both A. adenophora and A. riparia as widely distributed weeds
Answer: That's right. The sentence was modified as 'Although native to Mexico, it was first observed in China in the 20th century [26,27]'.
In the introduction there is no mention that the bio-fumigant plant used has been recorded with fungicidal effects in its leaves (due to essential oils), a reason for choosing this species, and only later in the Discussion (line 279-283) is it mentioned. This would probably explain why the leaves are more effective than the root-stems.
Answer: The present research has a great innovation. The purpose of this research was to study the bio-fumigant property of E. adenophorum in soil-borne pathogen control included Pythium, Fusarium and nematode, which was mentioned in the introduction about the ability of E. adenophorum bio-fumigant. With your permission, dear reviewer, the research and study on the bio-fumigant effects of E. adenophorum in controlling the nematode M. incognita, which was published in 2023 in Agriculture journal, was added in the introduction.
“The dried root-stems of E. adenophorum Spreng as a bio-fumigant plant had control effects on nematode Meloidogyne incognita in cucumber cultivar (Jinyou 35) greenhouse in China” (Parsiaaref et al., 2023).
Line 260 (Fig. 4) and line 264 (Fig. 5): The figures are not stand alone, need to explain the clustering (PCA?) and identify the figures without needing to consult the text.
Answer: The figures were modified. For a better understanding of the figures, the sentence “The dark blue color represents a correlation of +1, while the red color represents a correlation of -1” was added to the description of figure 4 and 5.
Line 387: There are no voucher specimens of the bio-fumigant plant used in the experiments and cited as deposited in a recognized herbarium, making later verification impossible.
Answer: The following sentence was added: The voucher specimen was deposited there with its scientific name.
Line 190: Table 3 heading time should be times also in line 208.
Answer: Modified as these
Line 247: supplemental should be supplementary.
Answer: Modified as these
Thank you again for your worthy comment, enhancing our manuscript.
Reviewer 2 Report
Comments and Suggestions for Authors
In this paper, the authors studied the antifungal effects of the plant Eupatorium adenophorum Spreng as a bio-fumigant plant, as such or in combination with biochar against the soil phytopathogenic fungus, Pythium aphanidermatum (Edson) Fitz. The paper is well structured, achieves its objectives, the documentation is extensive, and the experiments are carefully designed and performed. The comparative effect of different parts (leaves, stems-roots, leaves + biochar and stems-roots + biochar) of the Eupatorium adenophorum plant at different concentrations and at different time intervals was evaluated. The maximum antifungal effect was found, in the case of the leaves of the plant, at the maximum effective concentration of 79 g/kg soil. Also, the highest level of control, increased with time and reached 82.4% and 71% in 27 days of 30 and 60, respectively. The conclusions are consistent with the results presented by various analyzed studies, and the novelty is that E. adenophorum was studied as a dry plant for the control of the antifungal effect on the telluric fungus P. aphanidermatum. Tables and graphs support the statements in the text and help to better understand the results.
Author Response
In this paper, the authors studied the antifungal effects of the plant Eupatorium adenophorum Spreng as a bio-fumigant plant, as such or in combination with biochar against the soil phytopathogenic fungus, Pythium aphanidermatum (Edson) Fitz. The paper is well structured, achieves its objectives, the documentation is extensive, and the experiments are carefully designed and performed. The comparative effect of different parts (leaves, stems-roots, leaves + biochar and stems-roots + biochar) of the Eupatorium adenophorum plant at different concentrations and at different time intervals was evaluated. The maximum antifungal effect was found, in the case of the leaves of the plant, at the maximum effective concentration of 79 g/kg soil. Also, the highest level of control, increased with time and reached 82.4% and 71% in 27 days of 30 and 60, respectively. The conclusions are consistent with the results presented by various analyzed studies, and the novelty is that E. adenophorum was studied as a dry plant for the control of the antifungal effect on the telluric fungus P. aphanidermatum. Tables and graphs support the statements in the text and help to better understand the results.
Answer: Respected reviewer, thank you very much for your kind comment about the new manuscript. We hope that the results obtained from our research on the use of bio-fumigant plants to control Pythium can help researchers in the field of soil pathogens.
Round 2
Reviewer 1 Report
Comments and Suggestions for Authors
While there have been some improvements the article is still lacking to be accepted. The responses appear rather rushed.
In reference 26 the name Eupatorium adenophorum Spreng. is a synonym of Ageratina adenophora (Spreng.) R.M. King & H. Rob. and therefore, the correct name is Ageratina adenophora (Spreng.) R.M. King & H. Rob. which should be used. This is widely accepted in the botanical literature and online databases. Also, Spreng. has a full stop as it is shortened from Sprengel.
Line 190 paramters in title = parameters?
Figures improved however the clustering not explained in the legends. (stand-alone)
Line 392 Voucher specimen deposited (voucher number not given) there (herbarium acronym not given according to Index Herbariorum)..
Comments on the Quality of English LanguageSpelling of some English words were incorrect
Author Response
First Reviewer:
Comments: In reference 26 the name Eupatorium adenophorum Spreng. is a synonym of Ageratina adenophora (Spreng.) R.M. King & H. Rob. and therefore, the correct name is Ageratina adenophora (Spreng.) R.M. King & H. Rob. which should be used. This is widely accepted in the botanical literature and online databases. Also, Spreng. has a full stop as it is shortened from Sprengel.
Answer: Although the synonym of the scientific name of the Ageratina adenophora (Spreng.) R.M. King & H. Rob. plant is Eupatorium adenophorum Spreng.
In many recent studies, the synonym Eupatorium adenophorum Spreng has been used instead of Ageratina adenophora (Spreng.), such as:
Tripathi, Y.C., Saini, N., Anjum, N., Verma, P.K. A Review of Ethnomedicinal, Phytochemical, Pharmacological and Toxicological Aspects of Eupatorium adenophorum Spreng. Asian Journal of Biomedical and Pharmaceutical Sciences. 2019, 9(68):25-35
Abstract
Eupatorium adenophorum Spreng belonging to the family Asteraceae have traditionally been used as folklore medicine across the world. In traditional system of medicine, it is regarded as anti-inflammatory, antimicrobial, antiseptic, analgesic, antipyretic, blood and coagulant. The present review summarizes the updated information concerning the ethnomedicinal, phytochemical, pharmacological and toxicological aspects of E. adenophorum. A thorough bibliographic investigation was carried out by analyzing worldwide accepted scientific data base (Pub Med, Google Scholar, Scopus SciFinder, and Web of Science), thesis, recognized books and other accessible literature from 1980 to 2017. The phytochemical and pharmacological studies demonstrated that E. adenophorum possess a wide spectrum of pharmacological activities, such as anti-inflammatory, analgesic, antipyretic, antioxidant, antibacterial, antifungal, antitumor, antioxidant, antiseptic and cytotoxic activities which could be attributed to the presence of array of phytochemicals of various groups including terpenoids, phytosterols, alkaloids, flavonoids, phenolic acids, coumarins, phenylpropanoids, sesquiterpene lactones, polysaccharides, and essential oil. Modern phytochemical and pharmacological studies have led to the isolation and characterization of a number of bioactive compounds from different parts of the plant as well as validation of its traditional medicinal uses. However, certain known toxic effects of E. adenophorum demand a thorough study of long-term toxicity and other toxicological aspects.
Furthermore, the relationship of molecular structures of compounds of E. adenophorum with its various pharmacological activities needs further study and confirmation.
Keywords: Eupatorium adenophorum, Traditional uses, Phytochemistry, Pharmacology, Toxicity
Kundu, A.; Saha, S.; Walia, S.; Dutta, T.K. Antinemic Potentiality of chemical constituents of Eupatorium adenophorum Spreng leaves against Meloidogyne incognita. Natl. Acad. Sci. Lett. 2016, 39, 145–149.
Abstract
Volatile oil of Eupatorium adenophorum leaves was extracted and analyzed by gas chromatography–mass spectrometry (GC–MS). Sesquiterpenes were identified as major constituents in the essential oil. Four cadinene sesquiterpenes were isolated from ethyl acetate extract of leaves and identified spectroscopically. Chemical constituents were evaluated for antinemic activity against Meloidogyne incognita. Antinemic activity of volatile oil was in the range of LC50 133.7–189.2 µg ml−1. Methanolic concentrate exhibited maximum antinemic activity (LC50 93.7–162.4 µg ml−1) followed by hexane extract (LC50 99.8–127.2 µg ml−1). Among pure constituents, 5, 6-dihydroxycadinan-3-ene-2,7-dione exhibited significant activity (LC50 151.8 µg ml−1).
Damodar Singh, Y., Mukhopadhayay, S.K., Ayub Shah, M.A., Ayub Ali, M., Tolenkhomba, T.C. Pathology of Eupatorium adenophorum (Sticky snakeroot) toxicity in mice. International Multidisciplinary Research Journal 2012, 2(2):16-21
Abstract
The leaves of Eupatorium adenophorum Spreng were powdered and extracted with methanol. An acute oral toxicity study was conducted in male Swiss albino mice and a LD50 of 3501 mg/kg was obtained during 14 days observation period. Twenty Swiss albino mice (male) randomly divided into four groups were administered orally with vehicle (5% tween 80), 1/20th (i.e. 175 mg/kg), 1/10th (i.e. 350 mg/kg) and 1/5th (i.e. 750 mg/kg) LD50 doses of methanolic leaf extract of E. adenophorum Spreng; respectively for a period of 30 days. The mice were sacrificed on day-31 and the liver dissected out freed from adherent tissue weighed to nearest milligram. The liver histology, estimations of biochemical contents and enzyme activities were carried out. Treatment of the mice with methanolic extract of E. adenophorum at the dose level of 750 mg/kg (i.e. 1/5th LD50) elicited hepatotoxicity and the animals had yellow discoloration of liver, subcutaneous tissue and musculature indicating jaundice. Study on liver enzymes revealed marked increase in the activities of alkaline phosphatase (ALP), alanine transaminase (ALT), aspartate transaminase (AST) and lactate dehydrogenase (LDH), while significant increase in serum bilirubin level. Histopathological examination of the livers of the group IV animals had focal areas of necrosis and bile duct proliferation. Elevation in plasma bilirubin concomitant with alterations in enzyme profile and histopathological lesions are consistent with liver injury and cholestasis
Xiao-yu, S., Lu, Z., Wei-guo, S. Review on studies of Eupatorium adenophoruman important invasive species in China. Journal of Forestry Research. 2004, 15(4):319-322
Abstract
Eupatorium adenophorum Spreng. was introduced in Yunnan Province of China around 1940. Since then it has been spreading rapidly, particularly in the southern and southwestern parts of China and caused serious economic loss. The biological research and integrated control on E. adenophorum were carried out from 1980’s in Yunnan Province. Together with other 15 invasive external species, the weed has been listed in the White Paper by The State Environmental Protection Administration of China. This paper briefly reviews the studies on natural distribution, biological character, ecological character, chemical component, hazard, potential application and the control of E. adenophorum. The research direction for this invasive external species in future was also discussed.
Many studies have been conducted around the world and in China on the bio-fumigant plant Eupatorium adenophorum Spreng., and the scientific name Eupatorium adenophorum Spreng. has been used in all references.
The new manuscript, which contains a study of the effects of bio-fumigant and antifungal Eupatorium adenophorum Spreng against Pythium aphanidermatum (Edson) Fitz, is one of the research topics related to my doctoral thesis.
In 2023 and 2024, I published two articles in the Agriculture journal on the effects of bio-fumigant and nematicide Eupatorium adenophorum Spreng on the root-knot nematode Meloidogyne incognita.
“Parsiaaref, S.; Cao, A.; Li, Y.; Ebadollahi, A.; Parmoon, G.; Wang, Q.; Yan, D.; Fang, W.; Zhang, M. Nematicidal and toxicity effects of Eupatorium adenophorum Spreng against the root-knot nematode Meloidogyne incognita in soil producing cucumber. Agriculture. 2023, 13, 1109. https://doi.org/10.3390/agriculture13061109.”
“Parsiaaref, S., Cao, A., Li, Y., Ebadollahi, A., Parmoon, G., Wang, Q., Yan, D., Fang, W., Huang, B., Zhang, M. The Main Compounds of Bio-Fumigant Plants and Their Role in Controlling the Root-Knot Nematode Meloidogyne incognita (Kofoid and White) Chitwood. Agriculture 2024, 14(2), 261; https://doi.org/10.3390/agriculture14020261”
Because the new manuscript is a continuation of my doctoral thesis research, I ask the esteemed reviewer to allow me to use the scientific name Eupatorium adenophorum Spreng, which is a synonym of Ageratina adenophora (Spreng.) R.M. King & H. Rob. This way, a similar name is used in all my articles.
Comments: Line 190 paramters in title = parameters?
Answer: Modified as these
Comments: Figures improved however the clustering not explained in the legends. (stand-alone)
Answer: Modified as these
Comments: Line 392 Voucher specimen deposited (voucher number not given) there (herbarium acronym not given according to Index Herbariorum).
Answer: Regarding the respected reviewer's comment, I feel it necessary to introduce one of the researches conducted as a reference.
Sang, W., Zhu, L., Axmacher, J.C. Invasion pattern of Eupatorium adenophorum Spreng in southern China. Biological Invasions. 2010, 12(6):1721-1730. DOI:10.1007/s10530-009-9584-3
Abstract: This study provides a detailed analysis of the invasion by Eupatorium adenophorum Spreng (Crofton weed) from Burma and Vietnam into Southern China since the 1940s. Currently, E. adenophorum’s main colonisation area is located in theYunnan–Guizhou Plateau of China, where it has caused prominent economic and ecological problems. Sixty-three years ago, Crofton weed appeared in Menghai county, Yunnan Province, from where it dispersed northwards and eastwards at an average speed of 20 km year-1. The invasion of E. adenophorum nonetheless showed pronounced variations in both time and space. Spread was relatively slow in the initial invasion period between 1940 and 1950, while the most rapid range expansions occurred in the1980s. Environmental conditions at native and invaded sites were significantly different, reflecting a great adaptability of the species during colonization. These changes were greater than habitat differences between colonized and many adjacent un-colonized sites in Southern China. Therefore, immediate measures are required to stop a further northward and eastward expansion of Crofton weed.
Keywords: China, Crofton weed, Eupatorium adenophorum, Invasive species, Spatial distribution
Methods:
Distribution and spreading pattern of Croftonweed in China
Information on the distribution of invasive species both in native and colonized areas can be obtained by consulting data from herbarium collections (Chauvelet al. 2006). This information is widely available on the Internet and can be used for a wide range of ecological purposes (Graham et al. 2004). Despite suffering from several biases (Ponder et al. 2001; Delisle et al. 2003), herbarium specimens form an important source of information when reconstructing the introduction and colonization pattern of species (Weber 1998; Lambrinos 2001; Delisle et al. 2003;Stadler et al. 1998).
Records of the native range of E. adenophorum were obtained from the Global Biodiversity Infor-mation Facility (www.gbif.org). A total of 125 locations for Crofton weed occurring in native Mexico were identified. Each record included sampling location and habitat information within the species’ native distribution range in Mexico. These sites formed a representative selection of home-range locations situated in the central states of Guanajuato, Mexico, Morelos, Puebla, Veracruz, Hidalgo, and in the central southern states of Guerrero, Michoacan and Oaxaca (http://www.ars-grin.gov/cgi-bin/npgs/html/taxon.pl?316409, Central Mexico). The climatic and environmental characteristics of the native range locations were gained using GIS databases (ESRI 2003;http://www.cern.ac.cn/0index/;http://edcdaac.usgs.gov/gtopo30/;http://www.sage.wisc.edu/atlas/).
Records of E. adenophorum in its colonized range in China were collected using seven regional and national herbaria collections, while the extent of its current range was verified during field visits. Each specimen in the herbaria collections was checked for correct identification, collection number, location, date and relevant environmental factors. Specimens with ambiguous information were not used. The field sites surveyed were located in Yunnan, Sichuan, Guangxi, Guizhou, Hubei and Chongqing provinces. Due to its distinctive morphological characteristics, the plant is easily identified by local people. In China, it usually occurs in grasslands, forests, cropland and on road-sides. We visited sites near the invasion fronts as well as, following reports by local people, isolated sites beyond the front. Using both herbarium material covering dates from 1940 to 2003 and the additional field visits, the species could be traced back to a total of 441 localities. These localities were geo-referenced and fed into a GIS database to link them to prevailing environmental parameters at the respective sites.
The occurrence records were separated into a decadal time series, and the occurrence points were converted into county distributions, with counties representing areas between ca. 1,000 and 3,000 km2
A set of environmental parameters were chosen as indicators of the physical environment and recorded for each county, as well as for the sites at the species’ native home range. These included the following climatic factors: maximum, minimum and annual mean temperature, mean annual precipitation, pre-cipitation in the driest month and precipitation in the wettest month of the year for the period 1971–2000, using the Chinese Ecosystem Research Network dataset (http://www.cern.ac.cn/0index/). Furthermore, data on elevation and on soil pH, derived from the International Geosphere-Biosphere Program (IGBP)-DIS dataset (http://www.sage.wisc.edu/atlas/)wereusedin the analyses. Where necessary, we used Spline-interpolation of the environmental data to extract values for our geo-referenced sites. These eight environmental parameters were collected for both the 125 native sites identified in Mexico and all 411 colonized sites in China, as well as in an additional 441 randomly selected sites within a 100 km radius of the species’ current colonization fronts to compare the environmental conditions at these three sets of locations.
One of the researches that Professor Aocheng Cao and his colleagues conducted in the field of detection and control of Eupatorium adenophorum Spreng is described in the following reference:
Wang, C., Lin, H., Feng, Q., Jin, C., Cao, A., He, L. A New Strategy for the Prevention and Control of Eupatorium adenophorum under Climate Change in China. Sustainability. 2017, 9, 2037; doi:10.3390/su9112037
Abstract:
Eupatorium adenophorum has caused tremendous ecological and economic losses in China since the 1940s. Although a great deal of money has been expended on the prevention and control of the weed, the situation is still deteriorating. To identify its crucial environmental constraints, an ecological niche factor analysis was employed. The distribution of the weed was predicted by the maximum entropy model. The results indicate that the temperature in winter is more influential than that in other quarters of a year, and the maximum temperature in March restricts the spread of E. adenophorum most. Currently, the weed is mainly distributed in four provinces of southwest China. From the present to the 2080s, the center of L3, which has a potential distribution probability of 0.7 to 1.0, will move 53 km to the southwest. Accordingly, the area of L3 will expand by 16.04%. To prevent its further expansion, we suggest differentiating the prevention and control measures according to the potential distribution levels predicted. Meanwhile, the integration of various means of removal and comprehensive utilization of E. adenophorum is highly encouraged. Additionally, precautions should be taken in regions that have not yet, or have been only slightly, invaded by the E. adenophorum.
Keywords: species invasion; maximum entropy (MaxEnt); ecological niche model; ecological niche factor analysis; species distribution modeling
Round 3
Reviewer 1 Report
Comments and Suggestions for Authors
As a botanical journal I feel that current nomenclature should be followed and as the authors agree and cite reference 26 however they still cite the name throughout incorrectly. Sometimes in the text the author abreviation does not have a full stop as it is should according to botanical rules. No voucher specimens linked to herbaria are given for the material used in the experiments.
Author Response
Dear Editors,
Thank you very much for reviewing our manuscript. We are very grateful for the constructive and helpful comments of Reviewer 1. Reviewer 1 pointed out an important point. Since Plants journal is a botanical journal, reviewer 1's comment about changing the name of the plant Eupatorium adenophorum and writing about voucher specimens is appreciated and highly respected. We have considered all comments and modified the manuscript according to them. The corrections were highlighted with yellow color in the text. Please see the comments, our replies, and modifications as follows;
First Reviewer:
As a botanical journal I feel that current nomenclature should be followed and as the authors agree and cite reference 26 however they still cite the name throughout incorrectly. Sometimes in the text the author abreviation does not have a full stop as it is should according to botanical rules. No voucher specimens linked to herbaria are given for the material used in the experiments.
Answer: Thank you very much for your very nice comments. I am sure that writing the correct scientific name of plant and voucher specimens will increase and improve the quality of our manuscript. Your constructive comment increased my awareness of the bio-fumigant plant I used in my research.
Manuscript title: Studying the antifungal effects of Eupatorium adenophorum Spreng as a bio-fumigant plant apart and in combination with biochar against Pythium aphanidermatum (Edson) Fitz
Answer: Title changed to
Studying the antifungal effects of Ageratina adenophora (Sprengel) R. King & H. Robinson (=Eupatorium adenophorum Sprengel) as a bio-fumigant plant apart and in combination with biochar against Pythium aphanidermatum (Edson) Fitz
Line 24: Eupatorium adenophorum (Spreng)
Answer: Modified as these: Ageratina adenophora (Sprengel) R. King & H. Robinson (=Eupatorium adenophorum Sprengel)
Line 31: E. adenophorum
Answer: Modified as these: A. adenophora
Line 33: E. adenophorum
Answer: Modified as these: A. adenophora
Line 35: E. adenophorum
Answer: Modified as these: A. adenophora
Line 37: Eupatorium adenophorum
Answer: Modified as these: Ageratina adenophora
Line 97: E. adenophorum
Answer: Modified as these: A. adenophora
Line 99: E. adenophorum
Answer: Modified as these: A. adenophora
Line 100: E. adenophorum
Answer: Modified as these: A. adenophora
Line 102: E. adenophorum
Answer: Modified as these: A. adenophora
Line 105: E. adenophorum
Answer: Modified as these: A. adenophora
Line 120: Eupatorium adenophorum
Answer: Modified as these: Ageratina adenophora
Line 132: Eupatorium adenophorum
Answer: Modified as these: Ageratina adenophora
Line 145: E. adenophorum
Answer: Modified as these: A. adenophora
Line 150: E. adenophorum
Answer: Modified as these: A. adenophora
Line 157: Eupatorium adenophorum
Answer: Modified as these: Ageratina adenophora
Line 164: E. adenophorum
Answer: Modified as these: A. adenophora
Line 172: E. adenophorum
Answer: Modified as these: A. adenophora
Line 194: E. adenophorum
Answer: Modified as these: A. adenophora
Line 200: Eupatorium adenophorum
Answer: Modified as these: Ageratina adenophora
Line 206: E. adenophorum
Answer: Modified as these: A. adenophora
Line 222: E. adenophorum
Answer: Modified as these: A. adenophora
Line 224: E. adenophorum
Answer: Modified as these: A. adenophora
Line 237: Eupatorium adenophorum
Answer: Modified as these: Ageratina adenophora
Line 245: E. adenophorum
Answer: Modified as these: A. adenophora
Line 270: Eupatorium adenophorum
Answer: Modified as these: Ageratina adenophora
Line 278: Eupatorium adenophorum
Answer: Modified as these: Ageratina adenophora
Line 284: E. adenophorum
Answer: Modified as these: A. adenophora
Line 292: E. adenophorum
Answer: Modified as these: A. adenophora
Line 304: E. adenophorum
Answer: Modified as these: A. adenophora
Line 333: E. adenophorum
Answer: Modified as these: A. adenophora
Line 345: E. adenophorum
Answer: Modified as these: A. adenophora
Line 367: E. adenophorum
Answer: Modified as these: A. adenophora
Line 390: E. adenophorum
Answer: Modified as these: A. adenophora
Line 393: E. adenophorum
Answer: Modified as these: A. adenophora
Line 396: E. adenophorum
Answer: Modified as these: A. adenophora
Line 400: E. adenophorum
Answer: Modified as these: A. adenophora
Line 401: Eupatorium adenophorum
Answer: Modified as these: Ageratina adenophora
Line 403-405: No voucher specimens linked to herbaria are given for the material used in the experiments.
Answer: Information about the Ageratina adenophora voucher specimens has been added to the manuscript text in the Materials and Methods section “A voucher specimen (No. 20060712) was deposited at College of the Chemistry, Beijing Normal University, Beijing 100875, People’s Republic of China”.
Line 405: E. adenophorum
Answer: Modified as these: A. adenophora
Line 420: E. adenophorum
Answer: Modified as these: A. adenophora
Line 473: E. adenophorum
Answer: Modified as these: A. adenophora
Line 474: E. adenophorum
Answer: Modified as these: A. adenophora
Line 476: E. adenophorum
Answer: Modified as these: A. adenophora
Line 480: E. adenophorum
Answer: Modified as these: A. adenophora
Line 483: E. adenophorum
Answer: Modified as these: A. adenophora
Line 498: E. adenophorum
Answer: Modified as these: A. adenophora
Line 499: E. adenophorum
Answer: Modified as these: A. adenophora
Line 511: E. adenophorum
Answer: Modified as these: A. adenophora
Line 512: E. adenophorum
Answer: Modified as these: A. adenophora
Line 548: E. adenophorum
Answer: Modified as these: A. adenophora
Line 553: E. adenophorum
Answer: Modified as these: A. adenophora
Line 556: E. adenophorum
Answer: Modified as these: A. adenophora
Line 559: Eupatorium adenophorum
Answer: Modified as these: Ageratina adenophora
Line 560: Eupatorium adenophorum
Answer: Modified as these: Ageratina adenophora
Supplementary Table S1: Main compounds extracted of dried roots-stems from Eupatorium adenophorum with SPME at four times
Answer: Modified as these: Main compounds extracted of dried roots-stems from Ageratina adenophora with SPME at four times
Supplementary Table S2: Main compounds extracted of dried leaves from Eupatorium adenophorum with SPME at four times
Answer: Modified as these: Main compounds extracted of dried leaves from Ageratina adenophora with SPME at four times

Round 4
Reviewer 1 Report
Comments and Suggestions for Authors
The issues have been resolved